# Articulated Pose Estimation by a Graphical Model with Image Dependent Pairwise Relations

**Xianjie Chen**
University of California, Los Angeles
Los Angeles, CA 90024
cxj@ucla.edu

**Alan Yuille**
University of California, Los Angeles
Los Angeles, CA 90024
yuille@stat.ucla.edu

## Abstract

We present a method for estimating articulated human pose from a single static image based on a graphical model with novel pairwise relations that make adaptive use of local image measurements. More precisely, we specify a graphical model for human pose which exploits the fact the local image measurements can be used both to detect parts (or joints) and also to predict the spatial relationships between them (Image Dependent Pairwise Relations). These spatial relationships are represented by a mixture model. We use Deep Convolutional Neural Networks (DCNNs) to learn conditional probabilities for the presence of parts and their spatial relationships within image patches. Hence our model combines the representational flexibility of graphical models with the efficiency and statistical power of DCNNs. Our method significantly outperforms the state of the art methods on the LSP and FLIC datasets and also performs very well on the Buffy dataset without any training.

## 1 Introduction

Articulated pose estimation is one of the fundamental challenges in computer vision. Progress in this area can immediately be applied to important vision tasks such as human tracking [2], action recognition [25] and video analysis.

Most work on pose estimation has been based on graphical model [8, 6, 27, 1, 10, 2, 4]. The graph nodes represent the body parts (or joints), and the edges model the pairwise relationships between the parts. The score function, or energy, of the model contains unary terms at each node which capture the local appearance cues of the part, and pairwise terms defined at the edges which capture the local contextual relations between the parts. Recently, DeepPose [23] advocates modeling pose in a holistic manner and captures the full context of all body parts in a Deep Convolutional Neural Network (DCNN) [12] based regressor.

In this paper, we present a graphical model with image dependent pairwise relations (IDPRs). As illustrated in Figure 1, we can reliably predict the relative positions of a part's neighbors (as well as the presence of the part itself) by *only* observing the local image patch around it. So in our model the local image patches give input to both the unary and pairwise terms. This gives stronger pairwise terms because data independent relations are typically either too loose to be helpful or too strict to model highly variable poses.

Our approach requires us to have a method that can extract information about pairwise part relations, as well as part presence, from local image patches. We require this method to be efficient and to share features between different parts and part relationships. To do this, we train a DCNN to output

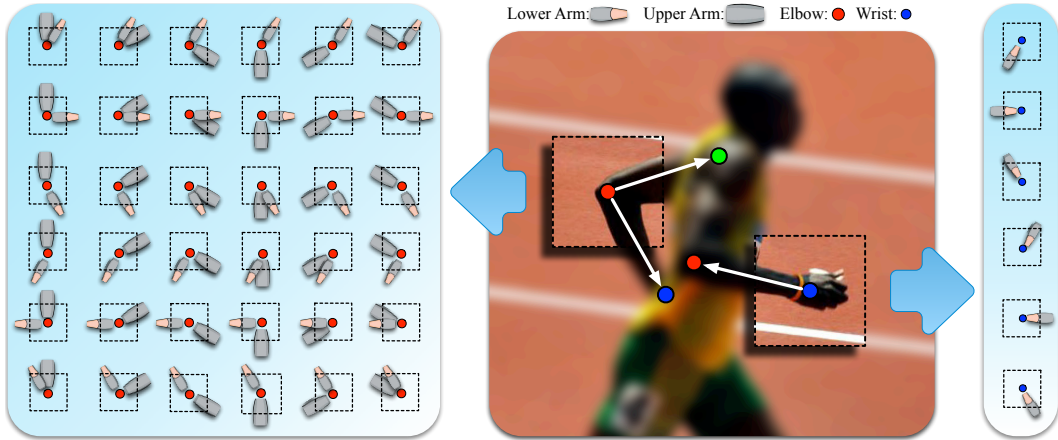

**Figure 1:** Motivation. The local image measurements around a part, *e.g.*, in an image patch, can reliably predict the relative positions of all its neighbors (as well as detect the part). Center Panel: The local image patch centered at the elbow can reliably predict the relative positions of the shoulder and wrist, and the local image patch centered at the wrist can reliably predict the relative position of the elbow. Left & Right Panels: We define different types of pairwise spatial relationships (*i.e.*, a mixture model) for each pair of neighboring parts. The Left Panel shows typical spatial relationships the elbow can have with its neighbors, *i.e.*, the shoulder and wrist. The Right Panel shows typical spatial relationships the wrist can have with its neighbor, *i.e.*, the elbow.

estimates for the part presence and spatial relationships which are used in our unary and pairwise terms of our score function. The weight parameters of different terms in the model are trained using Structured Supported Vector Machine (S-SVM) [24]. In summary, our model combines the representational flexibility of graphical models, including the ability to represent spatial relationships, with the data driven power of DCNNs.

We perform experiments on two standard pose estimation benchmarks: LSP dataset [10] and FLIC dataset [20]. Our method outperforms the state of the art methods by a significant margin on both datasets. We also do cross-dataset evaluation on Buffy dataset [7] (without training on this dataset) and obtain strong results which shows the ability of our model to generalize.

## 2  The Model

**The Graphical Model and its Variables:** We represent human pose by a graphical model $\mathcal{G} = (\mathcal{V}, \mathcal{E})$ where the nodes $\mathcal{V}$ specify the positions of the parts (or joints) and the edges $\mathcal{E}$ indicates which parts are spatially related. For simplicity, we impose that the graph structure forms a $K-$node tree, where $K = |\mathcal{V}|$. The positions of the parts are denoted by $\mathbf{l}$, where $\mathbf{l}_i = (x, y)$ specifies the pixel location of part $i$, for $i \in \{1, \ldots, K\}$. For each edge in the graph $(i, j) \in \mathcal{E}$, we specify a discrete set of spatial relationships indexed by $t_{ij}$, which corresponds to a mixture of different spatial relationships (see Figure 1). We denote the set of spatial relationships by $\mathbf{t} = \{t_{ij}, t_{ji} | (i, j) \in \mathcal{E}\}$. The image is written as $\mathbf{I}$. We will define a score function $F(\mathbf{l}, \mathbf{t} | \mathbf{t})$ as follows as a sum of unary and pairwise terms.

**Unary Terms:** The unary terms give local evidence for part $i \in \mathcal{V}$ to lie at location $\mathbf{l}_i$ and is based on the local image patch $\mathbf{I}(\mathbf{l}_i)$. They are of form:

$$U(\mathbf{l}_i | \mathbf{I}) = w_i \phi(i | \mathbf{I}(\mathbf{l}_i); \boldsymbol{\theta}), \tag{1}$$

where $\phi(. | .; \boldsymbol{\theta})$ is the (scalar-valued) appearance term with $\boldsymbol{\theta}$ as its parameters (specified in the next section), and $w_i$ is a scalar weight parameter.

**Image Dependent Pairwise Relational (IDPR) Terms:** These IDPR terms capture our intuition that neighboring parts $(i, j) \in \mathcal{E}$ can roughly predict their relative spatial positions using *only* local information (see Figure 1). In our model, the relative positions of parts $i$ and $j$ are discretized into several types $t_{ij} \in \{1, \ldots, T_{ij}\}$ (*i.e.*, a mixture of different relationships) with corresponding mean relative positions $\mathbf{r}_{ij}^{t_{ij}}$ plus small deformations which are modeled by the standard quadratic

deformation term. More formally, the pairwise relational score of each edge $(i, j) \in \mathcal{E}$ is given by:

$$
\begin{aligned}
R(\mathbf{l}_i, \mathbf{l}_j, t_{ij}, t_{ji} | \mathbf{I}) = & \quad \langle \mathbf{w}_{ij}^{t_{ij}}, \boldsymbol{\psi}(\mathbf{l}_j - \mathbf{l}_i - \mathbf{r}_{ij}^{t_{ij}}) \rangle + w_{ij} \varphi(t_{ij} | \mathbf{I}(\mathbf{l}_i); \boldsymbol{\theta}) \\
+ & \quad \langle \mathbf{w}_{ji}^{t_{ji}}, \boldsymbol{\psi}(\mathbf{l}_i - \mathbf{l}_j - \mathbf{r}_{ji}^{t_{ji}}) \rangle + w_{ji} \varphi(t_{ji} | \mathbf{I}(\mathbf{l}_j); \boldsymbol{\theta})
\end{aligned}
\tag{2}
$$

where $\boldsymbol{\psi}(\Delta \mathbf{l} = [\Delta x, \Delta y]) = [\Delta x \ \Delta x^2 \ \Delta y \ \Delta y^2]^\intercal$ are the standard quadratic deformation features, $\varphi(.|.; \boldsymbol{\theta})$ is the Image Dependent Pairwise Relational (IDPR) term with $\boldsymbol{\theta}$ as its parameters (specified in the next section), and $\mathbf{w}_{ij}^{t_{ij}}, w_{ij}, \mathbf{w}_{ji}^{t_{ji}}, w_{ji}$ are the weight parameters. The notation $\langle ., . \rangle$ specifies dot product and boldface indicates a vector.

**The Full Score:** The full score $F(\mathbf{l}, \mathbf{t} | \mathbf{I})$ is a function of the part locations $\mathbf{l}$, the pairwise relation types $\mathbf{t}$, and the input image $\mathbf{I}$. It is expressed as the sum of the unary and pairwise terms:

$$
F(\mathbf{l}, \mathbf{t} | \mathbf{I}) = \sum_{i \in \mathcal{V}} U(\mathbf{l}_i | \mathbf{I}) + \sum_{(i,j) \in \mathcal{E}} R(\mathbf{l}_i, \mathbf{l}_j, t_{ij}, t_{ji} | \mathbf{I}) + w_0,
\tag{3}
$$

where $w_0$ is a scalar weight on constant 1 (*i.e.*, the bias term).

The model consists of three sets of parameters: the mean relative positions $\mathbf{r} = \{\mathbf{r}_{ij}^{t_{ij}}, \mathbf{r}_{ji}^{t_{ji}} | (i, j) \in \mathcal{E}\}$ of different pairwise relation types; the parameters $\boldsymbol{\theta}$ of the appearance terms and IDPR terms; and the weight parameters $\mathbf{w}$ (*i.e.*, $w_i, \mathbf{w}_{ij}^{t_{ij}}, w_{ij}, \mathbf{w}_{ji}^{t_{ji}}, w_{ji}$ and $w_0$). See Section 4 for the learning of these parameters.

## 2.1 Image Dependent Terms and DCNNs

The appearance terms and IDPR terms depend on the image patches. In other words, a local image patch $\mathbf{I}(\mathbf{l}_i)$ not only gives evidence for the presence of a part $i$, but also about the relationship $t_{ij}$ between it and its neighbors $j \in \mathbb{N}(i)$, where $j \in \mathbb{N}(i)$ if, and only if, $(i, j) \in \mathcal{E}$. This requires us to learn distribution for the state variables $i, t_{ij}$ conditioned on the image patches $\mathbf{I}(\mathbf{l}_i)$. In order to specify this distribution we must define the state space more precisely, because the number of pairwise spatial relationships varies for different parts with different numbers of neighbors (see Figure 1), and we need also consider the possibility that the patch does not contain a part.

We define $c$ to be the random variable which denotes which part is present $c = i$ for $i \in \{1, ..., K\}$ or $c = 0$ if no part is present (*i.e.*, the background). We define $\mathbf{m}_{c\mathbb{N}(c)}$ to be the random variable that determines the spatial relation types of $c$ and takes values in $\mathbb{M}_{c\mathbb{N}(c)}$. If $c = i$ has one neighbor $j$ (*e.g.*, the wrist), then $\mathbb{M}_{i\mathbb{N}(i)} = \{1, \ldots, T_{ij}\}$. If $c = i$ has two neighbors $j$ and $k$ (*e.g.*, the elbow), then $\mathbb{M}_{i\mathbb{N}(i)} = \{1, \ldots, T_{ij}\} \times \{1, \ldots, T_{ik}\}$. If $c = 0$, then we define $\mathbb{M}_{0\mathbb{N}(0)} = \{0\}$.

The full space is represented as:

$$
\mathbb{S} = \cup_{c=0}^{K} \{c\} \times \mathbb{M}_{c\mathbb{N}(c)}
\tag{4}
$$

The size of the space is $|\mathbb{S}| = \sum_{c=0}^{K} |\mathbb{M}_{c\mathbb{N}(c)}|$. Each element in this space corresponds to a part with all the types of its pairwise relationships, or the background.

We use DCNN [12] to learn the conditional probability distribution $p(c, \mathbf{m}_{c\mathbb{N}(c)} | \mathbf{I}(\mathbf{l}_i); \boldsymbol{\theta})$. DCNN is suitable for this task because it is very efficient and enables us to share features. See section 4 for more details.

We specify the appearance terms $\phi(.|.; \boldsymbol{\theta})$ and IDPR terms $\varphi(.|.; \boldsymbol{\theta})$ in terms of $p(c, \mathbf{m}_{c\mathbb{N}(c)} | \mathbf{I}(\mathbf{l}_i); \boldsymbol{\theta})$ by marginalization:

$$
\begin{aligned}
\phi(i | \mathbf{I}(\mathbf{l}_i); \boldsymbol{\theta}) &= \log(p(c = i | \mathbf{I}(\mathbf{l}_i); \boldsymbol{\theta})) & (5) \\
\varphi(t_{ij} | \mathbf{I}(\mathbf{l}_i); \boldsymbol{\theta}) &= \log(p(m_{ij} = t_{ij} | c = i, \mathbf{I}(\mathbf{l}_i); \boldsymbol{\theta})) & (6)
\end{aligned}
$$

## 2.2 Relationship to other models

We now briefly discuss how our method relates to standard models.

**Pictorial Structure:** We recover pictorial structure models [6] by only allowing one relationship type (*i.e.*, $T_{ij} = 1$). In this case, our IDPR term conveys no information. Our model reduces to

standard unary and (image independent) pairwise terms. The only slight difference is that we use DCNN to learn the unary terms instead of using HOG filters.

**Mixtures-of-parts:** [27] describes a model with a mixture of templates for each part, where each template is called a "type" of the part. The "type" of each part is defined by its relative position with respect to its parent. This can be obtained by restricting each part in our model to only predict the relative position of its parent (*i.e.*, $T_{ij} = 1$, if $j$ is not parent of $i$). In this case, each part is associated with only one informative IDPR term, which can be merged with the appearance term of each part to define different "types" of part in [27]. Also this method does not use DCNNs.

**Conditional Random Fields (CRFs):** Our model is also related to the conditional random field literature on data-dependent priors [18, 13, 15, 19]. The data-dependent priors and unary terms are typically modeled separately in the CRFs. In this paper, we efficiently model all the image dependent terms (i.e. unary terms and IDPR terms) together in a single DCNN by exploiting the fact the local image measurements are reliable for predicting both the presence of a part and the pairwise relationships of a part with its neighbors.

## 3   Inference

To detect the optimal configuration for each person, we search for the configurations of the locations **l** and types **t** that maximize the score function: $(\mathbf{l}^*, \mathbf{t}^*) = \arg\max_{\mathbf{l},\mathbf{t}} F(\mathbf{l}, \mathbf{t}|\mathbf{I})$. Since our relational graph is a tree, this can be done efficiently via dynamic programming.

Let $\mathbb{K}(i)$ be the set of children of part $i$ in the graph ($\mathbb{K}(i) = \emptyset$, if part $i$ is a leaf), and $S_i(\mathbf{l}_i|\mathbf{I})$ be maximum score of the subtree rooted at part $i$ with part $i$ located at $\mathbf{l}_i$. The maximum score of each subtree can be computed as follow:

$$S_i(\mathbf{l}_i|\mathbf{I}) = U(\mathbf{l}_i|\mathbf{I}) + \sum_{k \in \mathbb{K}(i)} \max_{\mathbf{l}_k, t_{ik}, t_{ki}} \left( R(\mathbf{l}_i, \mathbf{l}_k, t_{ik}, t_{ki}|\mathbf{I}) + S_k(\mathbf{l}_k|\mathbf{I}) \right) \tag{7}$$

Using Equation 7, we can recursively compute the overall best score of the model, and the optimal configuration of locations and types can be recovered by the standard backward pass of dynamic programming.

**Computation:** Since our pairwise term is a quadratic function of locations, $\mathbf{l}_i$ and $\mathbf{l}_j$, the max operation over $\mathbf{l}_k$ in Equation 7 can be accelerated by using the generalized distance transforms [6]. The resulting approach is very efficient, taking $O(T^2 LK)$ time once the image dependent terms are computed, where $T$ is the number of relation types, L is the total number of locations, and K is the total number of parts in the model. This analysis assumes that all the pairwise spatial relationships have the same number of types, *i.e.*, $T_{ij} = T_{ji} = T, \forall (i,j) \in \mathcal{E}$.

The computation of the image dependent terms is also efficient. They are computed over all the locations by a single DCNN. Applying DCNN in a sliding fashion is inherently efficient, since the computations common to overlapping regions are naturally shared [22].

## 4   Learning

Now we consider the problem of learning the model parameters from images with labeled part locations, which is the data available in most of the human pose datasets [17, 7, 10, 20]. We derive type labels $t_{ij}$ from part location annotations and adopt a supervised approach to learn the model.

Our model consists of three sets of parameters: the mean relative positions $\mathbf{r}$ of different pairwise relation types; the parameters $\boldsymbol{\theta}$ of the image dependent terms; and the weight parameters $\mathbf{w}$. They are learnt separately by the K-means algorithm for $\mathbf{r}$, DCNN for $\boldsymbol{\theta}$, and S-SVM for $\mathbf{w}$.

**Mean Relative Positions and Type Labels:** Given the labeled positive images $\{(\mathbf{I}^n, \mathbf{l}^n)\}_{n=1}^N$, let $\mathbf{d}_{ij}$ be the relative position from part $i$ to its neighbor $j$. We cluster the relative positions over the training set $\{\mathbf{d}_{ij}^n\}_{n=1}^N$ to get $T_{ij}$ clusters (in the experiments $T_{ij} = 11$ for all pairwise relations). Each cluster corresponds to a set of instances of part $i$ that share similar spatial relationship with its neighbor part $j$. Thus we define each cluster as a pairwise relation type $t_{ij}$ from part $i$ to $j$ in our model, and use the center of each cluster as the mean relative position $\mathbf{r}_{ij}^{t_{ij}}$ associated with each

type. In this way, the mean relative positions of different pairwise relation types are learnt, and the type label $t_{ij}^n$ for each training instance is derived based on its cluster index. We use K-means in our experiments by setting $K = T_{ij}$ to do the clustering.

**Parameters of Image Dependent Terms:** After deriving type labels, each local image patch $\mathbf{I}(\mathbf{l}^n)$ centered at an annotated part location is labeled with category label $c^n \in \{1, \ldots, K\}$, that indicates which part is present, and also the type labels $\mathbf{m}_{c^n \mathbb{N}(c^n)}^n$ that indicate its relation types with all its neighbors. In this way, we get a set of labelled patches $\{\mathbf{I}(\mathbf{l}^n), c^n, \mathbf{m}_{c^n \mathbb{N}(c^n)}^n\}_{n=1}^{KN}$ from positive images (each positive image provides $K$ part patches), and also a set of background patches $\{\mathbf{I}(\mathbf{l}^n), 0, 0\}$ sampled from negative images.

Given the labelled part patches and background patches, we train a multi-class DCNN classifier by standard stochastic gradient descent using softmax loss. The DCNN consists of five convolutional layers, 2 max-pooling layers and three fully-connected layers with a final $|\mathbb{S}|$ dimensions softmax output, which is defined as our conditional probability distribution, *i.e.*, $p(c, \mathbf{m}_{c\mathbb{N}(c)}|\mathbf{I}(\mathbf{l}_i); \boldsymbol{\theta})$. The architecture of our network is summarized in Figure 2.

**Weight Parameters:** Each pose in the positive image is now labeled with annotated part locations and derived type labels: $(\mathbf{I}^n, \mathbf{l}^n, \mathbf{t}^n)$. We use S-SVM to learn the weight parameters $\mathbf{w}$. The structure prediction problem is simplified by using $0 - 1$ loss, that is all the training examples either have all dimensions of its labels correct or all dimensions of its labels wrong. We denote the former ones as *pos* examples, and the later ones as *neg* examples. Since the full score function (Equation 3) is linear in the weight parameters $\mathbf{w}$, we write the optimization function as:

$$\min_{\mathbf{w}} \frac{1}{2}\langle \mathbf{w}, \mathbf{w} \rangle + C \sum_n \max(0, 1 - y_n \langle \mathbf{w}, \boldsymbol{\Phi}(\mathbf{I}^n, \mathbf{l}^n, \mathbf{t}^n) \rangle), \tag{8}$$

where $y_n \in \{1, -1\}$, and $\boldsymbol{\Phi}(\mathbf{I}^n, \mathbf{l}^n, \mathbf{t}^n)$ is a sparse feature vector representing the $n$-th example and is the concatenation of the image dependent terms (calculated from the learnt DCNN), spatial deformation features, and constant 1. Here $y_n = 1$ if $n \in pos$, and $y_n = -1$ if $n \in neg$.

## 5 Experiment

This section introduces the datasets, clarifies the evaluation metrics, describes our experimental setup, presents comparative evaluation results and gives diagnostic experiments.

### 5.1 Datasets and Evaluation Metrics

We perform our experiments on two publicly available human pose estimation benchmarks: (i) the "Leeds Sports Poses" (LSP) dataset [10], that contains 1000 training and 1000 testing images from sport activities with annotated full-body human poses; (ii) the "Frames Labeled In Cinema" (FLIC) dataset [20] that contains 3987 training and 1016 testing images from Hollywood movies with annotated upper-body human poses. We follow previous work and use the observer-centric annotations on both benchmarks. To train our models, we also use the negative training images from the INRIAPerson dataset [3] (These images do not contain people).

We use the most popular evaluation metrics to allow comparison with previous work. Percentage of Correct Parts (PCP) is the standard evaluation metric on several benchmarks including the LSP dataset. However, as discussed in [27], there are several alternative interpretations of PCP that can lead to severely different results. In our paper, we use the stricter version unless otherwise stated, that is we evaluate only a *single* highest-scoring estimation result for one test image and a body part is considered as correct if *both* of its segment endpoints (joints) lie within 50% of the length of the ground-truth annotated endpoints (Each test image on the LSP dataset contains only one annotated person). We refer to this version of PCP as *strict* PCP.

On the FLIC dataset, we use both *strict* PCP and the evaluation metric specified with it [20]: Percentage of Detected Joints (PDJ). PDJ measures the performance using a curve of the percentage of correctly localized joints by varying localization precision threshold. The localization precision threshold is normalized by the scale (defined as distance between left shoulder and right hip) of each ground truth pose to make it scale invariant. There are multiple people in the FLIC images, so each

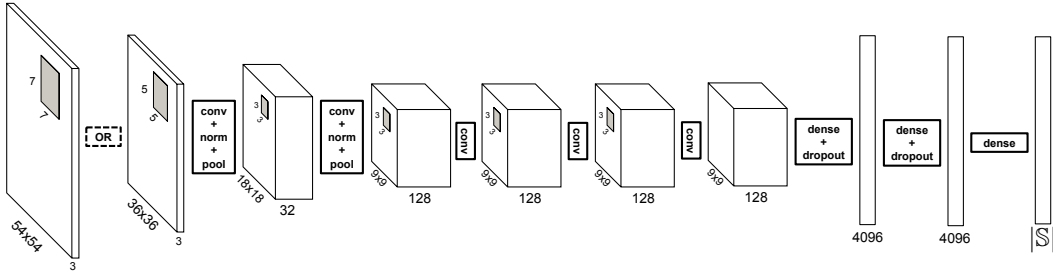

**Figure 2:** Architectures of our DCNNs. The size of input patch is $36 \times 36$ pixels on the LSP dataset, and $54 \times 54$ pixels on the FLIC dataset. The DCNNs consist of five convolutional layers, 2 max-pooling layers and three fully-connected (dense) layers with a final $|\mathbb{S}|$ dimensions output. We use dropout, local response normalization (norm) and overlapping pooling (pool) described in [12].

ground truth person is also annotated with a torso detection box. During evaluation, we return a *single* highest-scoring estimation result for each ground truth person by restricting our neck part to be localized inside a window defined by the provided torso box.

## 5.2  Implementation detail

**Data Augmentation:** Our DCNN has millions of parameters, while only several thousand of training images are available. In order to reduce overfitting, we augment the training data by rotating the positive training images through $360°$. These images are also horizontally flipped to double the training images. This increases the number of training examples of body parts with different spatial relationships with its neighbors (See the elbows along the diagonal of the Left Panel in Figure 1). We hold out random positive images as a validation set for the DCNN training. Also the weight parameters **w** are trained on this held out set to reduce overfitting to training data.

Note that our DCNN is trained using local part patches and background patches instead of the whole images. This naturally increases the number of training examples by a factor of $K$ (the number of parts). Although the number of dimensions of the DCNN final output also increases linearly with the number of parts, the number of parameters only slightly increase in the last fully-connected layer. This is because most of the parameters are shared between different parts, and thus we can benefit from larger $K$ by having more training examples per parameter. In our experiments, we increase $K$ by adding the midway points between annotated parts, which results in 26 parts on the LSP dataset and 18 parts on the FLIC dataset. Covering a person by more parts also reduces the distance between neighboring parts, which is good for modeling foreshortening [27].

**Graph Structure:** We define a full-body graph structure for the LSP dataset, and a upper-body graph structure for the FLIC dataset respectively. The graph connects the annotated parts and their midways points to form a tree (See the skeletons in Figure 5 for clarification).

**Settings:** We use the same number of types for all pairs of neighbors for simplicity. We set it as 11 on all datasets (*i.e.*, $T_{ij} = T_{ji} = 11, \forall(i,j) \in \mathcal{E}$), which results in 11 spatial relation types for the parts with one neighbor (*e.g.*, the wrist), $11^2$ spatial relation types for the parts with two neighbors (*e.g.*, the elbow), and so forth (recall Figure 1). The patch size of each part is set as $36 \times 36$ pixels on the LSP dataset, and $54 \times 54$ pixels on the FLIC dataset, as the FLIC images are of higher resolution. We use similar DCNN architectures on both datasets, which differ in the first layer because of different input patch sizes. Figure 2 visualizes the architectures we used. We use the Caffe [9] implementation of DCNN in our experiments.

## 5.3  Benchmark results

We show *strict* PCP results on the LSP dataset in Table 1, and on the FLIC dataset in Table 2. We also show PDJ results on the FLIC dataset in Figure 3. As is shown, our method outperforms state of the art methods by a significant margin on both datasets (see the captions for detailed analysis). Figure 5 shows some estimation examples on the LSP and FLIC datasets.

| Method | Torso | Head | U.arms | L.arms | U.legs | L.legs | Mean |
|---|---|---|---|---|---|---|---|
| **Ours** | **92.7** | **87.8** | **69.2** | **55.4** | **82.9** | **77.0** | **75.0** |
| Pishchulin et al. [16] | 88.7 | 85.6 | 61.5 | 44.9 | 78.8 | 73.4 | 69.2 |
| Ouyang et al. [14] | 85.8 | 83.1 | 63.3 | 46.6 | 76.5 | 72.2 | 68.6 |
| DeepPose* [23] | - | - | 56 | 38 | 77 | 71 | - |
| Pishchulin et al. [15] | 87.5 | 78.1 | 54.2 | 33.9 | 75.7 | 68.0 | 62.9 |
| Eichner&Ferrari [4] | 86.2 | 80.1 | 56.5 | 37.4 | 74.3 | 69.3 | 64.3 |
| Yang&Ramanan [26] | 84.1 | 77.1 | 52.5 | 35.9 | 69.5 | 65.6 | 60.8 |

**Table 1:** Comparison of *strict* PCP results on the LSP dataset. Our method improves on all parts by a significant margin, and outperforms the best previously published result [16] by 5.8% on average. Note that DeepPose uses Person-Centric annotations and is trained with an extra 10,000 images.

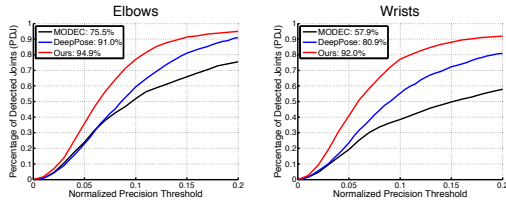

| Method | U.arms | L.arms | Mean |
|---|---|---|---|
| **Ours** | **97.0** | **86.8** | **91.9** |
| MODEC[20] | 84.4 | 52.1 | 68.3 |

**Table 2:** Comparison of *strict* PCP results on the FLIC dataset. Our method significantly outperforms MODEC [20].

**Figure 3:** Comparison of PDJ curves of elbows and wrists on the FLIC dataset. The legend shows the PDJ numbers at the threshold of 0.2.

## 5.4 Diagnostic Experiments

We perform diagnostic experiments to show the cross-dataset generalization ability of our model, and better understand the influence of each term in our model.

**Cross-dataset Generalization:** We directly apply the trained model on the FLIC dataset to the official test set of Buffy dataset [7] (*i.e.*, no training on the Buffy dataset), which also contains upper-body human poses. The Buffy test set includes a subset of people whose upper-body can be detected. We get the newest detection windows from [5], and compare our results to previously published work on this subset.

Most previous work was evaluated with the official evaluation toolkit of Buffy, which uses a less strict PCP implementation[1]. We refer to this version of PCP as *Buffy* PCP and report it along with the *strict* PCP in Table 3. We also show the PDJ curves in Figure 4. As is shown by both criterions, our method significantly outperforms the state of the arts, which shows the good generalization ability of our method. Also note that both DeepPose [23] and our method are trained on the FLIC dataset. Compared with Figure 3, the margin between our method and DeepPose significantly increases in Figure 4, which implies that our model generalizes better to the Buffy dataset.

| Method | U.arms | L.arms | Mean |
|---|---|---|---|
| Ours* | 96.8 | 89.0 | 92.9 |
| Ours* *strict* | 94.5 | 84.1 | 89.3 |
| Yang[27] | 97.8 | 68.6 | 83.2 |
| Yang[27] *strict* | 94.3 | 57.5 | 75.9 |
| Sapp[21] | 95.3 | 63.0 | 79.2 |
| FLPM[11] | 93.2 | 60.6 | 76.9 |
| Eichner[5] | 93.2 | 60.3 | 76.8 |

**Table 3:** Cross-dataset PCP results on Buffy test subset. The PCP numbers are *Buffy* PCP unless otherwise stated. Note that our method is trained on the FLIC dataset.

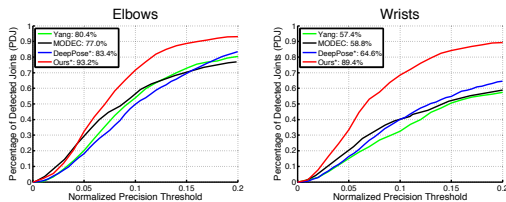

**Figure 4:** Cross-dataset PDJ curves on Buffy test subset. The legend shows the PDJ numbers at the threshold of 0.2. Note that both our method and DeepPose [23] are trained on the FLIC dataset.

| Method | Torso | Head | U.arms | L.arms | U.legs | L.legs | Mean |
|---|---|---|---|---|---|---|---|
| *Unary-Only* | 56.3 | 66.4 | 28.9 | 15.5 | 50.8 | 45.9 | 40.5 |
| *No-IDPRs* | 87.4 | 74.8 | 60.7 | 43.0 | 73.2 | 65.1 | 64.6 |
| Full Model | **92.7** | **87.8** | **69.2** | **55.4** | **82.9** | **77.0** | **75.0** |

**Table 4:** Diagnostic term analysis *strict* PCP results on the LSP dataset. The unary term alone is still not powerful enough to get good results, even though it's trained by a DCNN classifier. *No-IDPRs* method, whose pairwise terms are not dependent on the image (see Terms Analysis in Section 5.4), can get comparable performance with the state-of-the-art, and adding IDPR terms significantly boost our final performance to 75.0%.

**Terms Analysis:** We design two experiments to better understand the influence of each term in our model. In the first experiment, we use only the unary terms and thus all the parts are localized independently. In the second experiment, we replace the IDPR terms with image independent priors (*i.e.*, in Equation 2, $w_{ij}\varphi(t_{ij}|\mathbf{I}(\mathbf{l}_i);\boldsymbol{\theta})$ and $w_{ji}\varphi(t_{ji}|\mathbf{I}(\mathbf{l}_j);\boldsymbol{\theta})$ are replaced with scalar prior terms $b_{ij}^{t_{ij}}$ and $b_{ji}^{t_{ji}}$ respectively), and retrain the weight parameters along with the new prior terms. In this case, our pairwise relational terms do not depend on the image, but instead is a mixture of Gaussian deformations with image independent biases. We refer to the first experiment as *Unary-Only* and the second one as *No-IDPRs*, short for No IDPR terms. The experiments are done on the LSP dataset using identical appearance terms for fair comparison. We show *strict* PCP results in Table 4. As is shown, all terms in our model significantly improve the performance (see the caption for detail).

## 6   Conclusion

We have presented a graphical model for human pose which exploits the fact the local image measurements can be used both to detect parts (or joints) and also to predict the spatial relationships between them (Image Dependent Pairwise Relations). These spatial relationships are represented by a mixture model over types of spatial relationships. We use DCNNs to learn conditional probabilities for the presence of parts and their spatial relationships within image patches. Hence our model combines the representational flexibility of graphical models with the efficiency and statistical power of DCNNs. Our method outperforms the state of the art methods on the LSP and FLIC datasets and also performs very well on the Buffy dataset without any training.

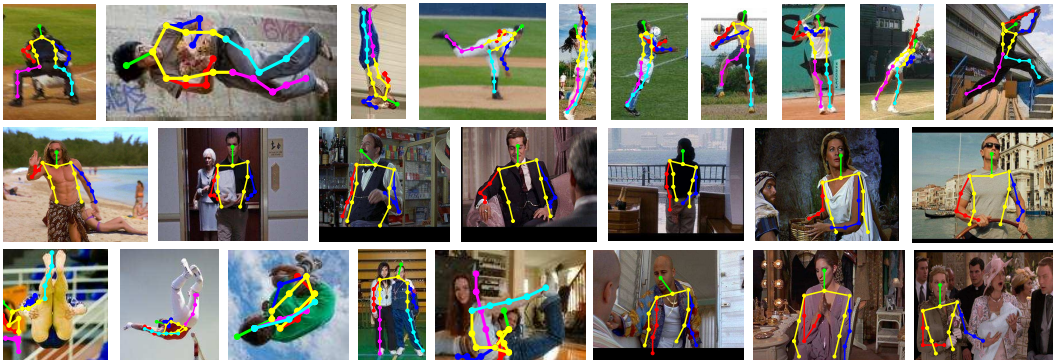

**Figure 5:** Results on the LSP and FLIC datasets. We show the part localization results along with the graph skeleton we used in the model. The last row shows some failure cases, which are typically due to large foreshortening, occlusions and distractions from clothing or overlapping people.

## 7   Acknowledgements

This research has been supported by grants ONR MURI N000014-10-1-0933, ONR N00014-12-1-0883 and ARO 62250-CS. The GPUs used in this research were generously donated by the NVIDIA Corporation.

## Footnotes

[1]A part is considered correctly localized if the *average* distance between its endpoints (joints) and ground-truth is less than 50% of the length of the ground-truth annotated endpoints.

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
