[Reviews · NeurIPS 2014]

Submitted by Assigned_Reviewer_13

The paper proposes a graphical model for human pose estimation in still images, where the nodes correspond to part locations, and edges encode the spatial relationships between parts. The main contribution of the paper is the design of the pairwise potentials, which is based on clustering the relative spatial locations of parts, and learning a ConvNet to predict the clusters (spatial configuration types) from image patches. The unary potentials encode the likelihood of a part appearing at a particular location, which is also learned using a patch-based ConvNet. The method is evaluated on several datasets (FLIC, LSP, and Buffy), where it achieves the state of the art, outperforming holistic DeepPose prediction model [17]. It is also shown that the proposed image-dependent pairwise terms are indeed important, as they lead to better performance than image-independent priors. The paper is well written and sufficiently detailed.

Questions for the authors:
1) How does the choice of the negative image dataset (l.244-245) affect the results?
2) Training set augmentation by rotation increases the training set size, but produces unrealistic images; how does the performance change when this augmentation is switched off?
3) Is it possible to jointly optimize the weight parameters and the image-dependent terms (ConvNet models)?
Summary: The paper introduces a graphical model for pose estimation, which incorporates novel image-dependent pairwise terms. The proposed method achieves the state of the art results.

Submitted by Assigned_Reviewer_16

This work researches the human pose estimation problem and takes the traditional approach of graphical models. The key idea of this work is to improve the pairwise term in the graphical model to make it dependent on image data. This is motivated by the observation that the relative position between two parts can be inferred from the local image patches around the parts. To model the relationship between relative position and local image patches, this work trains a multi-class deep convnet, where each class corresponds to a given part having a specific relation type with its neighboring part(s). Experimental study is conducted on benchmark data sets for human pose estimation and the advantage of this work is well demonstrated.

Pros:
1. This work achieves clear improvement over existing works, including the DeepPose newly published in CVPR2014. The improvement can be seen on each component and the magnitude is significant.
2. The use of deep network to infer the relative position between parts from local image patches is new and this demonstrates again the power of convnets in vision tasks.

Cons:
1. The idea of changing a data-independent pairwise term to a data-dependent one is not new and has been widely used in pose estimation and image segmentation. However, it seems this work uses this idea very well.
2. The discussion of the relationship to existing methods in Section 2.2 shall be well expanded to include more pose estimation methods. In particular, how is the proposed idea related to and different from the "Poselet dependent pairwise terms" in ref.[11], Sec.3.3?

Detailed comments:
1. How many classes are used in the DCNN training, that is, the value of |S|?
2. On what extent is the improvement due to the use of DCNN? Say, if HOG is used to represent local image patches and then a multi-class SVM is trained, how much improvement can be expected?
3. All the examples shown in the left most of Figure 1 seem to be from the frontal view. In this work, has the variation of relative position w.r.t view point been explicitly handled? If yes, how is it handled?
Summary: The key idea is not fundamentally new in pose estimation. However, the idea is realized in a new and effective way via deep network and it achieves significant improvement over the state of the art.

Submitted by Assigned_Reviewer_31

This work proposes a method for monocular human pose estimation. The
proposed method is based on a tree-structured graphical model with
image dependent pairwise relationships. These relationships are
represented by a mixture model and local image measurements are used
to predict a mixture component for each pairwise term. Prediction is
based on features extracted from image patches using Deep
Convolutional Neural Networks (DCNN). DCNN are trained to predict
conditional probabilities for 1) spatial relations between body parts
and 2) presence of individual body parts. Finally, mixture weights
unary and pairwise terms are jointly learned in a max-margin
framework.

This is an interesting paper with several strong points:

- significance. This paper addresses a challenging and highly research problem of
monocular human pose estimation and thus it may have an impact on
significant part of community

- model. Using deep learning features for local conditioning of
pairwise terms in a graphical model is an interesting idea.

- experimental evaluation. The work provides some insights by breaking
down the performance of individual model components. The importance
of image conditioned pairwise terms for pose estimation performance
is evident.

- results. This work shows promising results significantly
improving over state of the art on standard pose estimation benchmarks.

However, this paper also has several weak points:

- conditioning of pairwise terms in a graphical model for human pose
estimation has been done before by [SJT10] (see citation below) and
[11]. I struggle to see the principal differences between this
method and the previous work, except that this paper uses
stronger image features and local conditioning. Moreover, it is not
clear that local conditioning of pairwise terms has more advantages
over global conditioning, as local conditioning allows to model
pairwise relationships between adjacent body parts *only*, which is
in contrast to the previous method [11] which models longer range
part dependencies.

- the proposed method heavily builds on a Yang&Ramanan (Y&R) model
[20,21] and reads like an extension of Y&R model with
image-conditioned pairwise terms.

- choosing a number of mixture components for the pairwise terms (11)
is not justified. Y&R model [21] uses 6 components only. Fairer
comparison between two models would be when the number of mixture
components is the same.

- using 0-1 loss during training is suboptimal, as at test time
different types of losses measuring the fraction of correctly
predicted body parts/joints are used. A more proper way would be to
use the same loss during testing and training.

- comparison to Deep Pose [17] on LSP dataset is not completely fair,
as [17] report results using person-centric annotations, which is a
harder task.

- some citations are missing

[SJT10] B. Sapp, C. Jordan, and B. Taskar. Adaptive pose priors for
pictorial structures. In CVPR’10.

[KU12] L. Karlinsky, and U. Shimon. Using linking features in learning
non-parametric part models. In ECCV'12

- unrealistic poses resulting from data augmentation by rotating each
positive training image through 360 with a step of 10 degrees may
hurt the performance. The paper advocates this procedure by saying
that the patches extracted from the unrealistically rotated poses
still contain information about local part relations. However, it
has been suggested [20,21] that the appearance of body parts is
rotation dependent, and thus is characteristic for particular part
orientations. Moreover, the local patches seem to be large enough to
include a significant portion of the context, e.g. supporting
surfaces for legs in upright or for arms in upside down positions,
which may enhance discriminative power.

Minor comments:

- saying that the method "performs very well on Buddy dataset without
any training" is a bit misleading. What is actually meant is that
the method was not trained on Buffy dataset, but trained on other
datasets and tested on Buffy.

- what is the training/testing time of the method?
Summary: This is an interesting way of using deep learning to locally condition
pairwise terms. However, principal differences to the previous work
are unclear.
Author Feedback
Author rebuttal: We thank the reviewers for their insightful comments.

@R13:
#How does the choice of the negative images affect the results?#
We use the same negative images for both the LSP and FLIC dataset. Using more carefully chosen negative images may further improve the performance as observed in [3].

#How does the performance change when training set augmentation is switched off?#
We use data augmentation since the number of training images is limited. Without the augmentation, our DCNN suffers from overfitting (and thus inferior performance), especially on the LSP dataset, which has fewer images (1000 images) and more varied poses.

#Is it possible to jointly optimize the weight parameters and the image-dependent terms?#
Training them jointly is not trivial, but is definitely worth exploring.

@R16:
#More discussion of the relationship with other methods, in particular with [11].#
We will add more discussion. In [11], all the pairwise terms are conditioned on the same mid-level representation of the holistic image (i.e. the poselet responses at different image positions). Intuitively, different region of the image should be predictive for different pairs of body parts. Conditioning on the same statistics from the holistic image introduces unnecessary nuisance variables for each pairwise term, which makes the learning harder and tend to overfit.

In our model, each pairwise term is conditioned on the predictive local image measurements instead of the holistic image. Other differences include that we use DCNNs and we improve performance.

#How many classes are used in the DCNN training, that is, the value of |S|?#
In the experiments, |S|=6227 on the LSP dataset, and |S|=5259 on the FLIC dataset.

#On what extent is the improvement due to the use of DCNN? Say, if HOG is used to represent local image patches and then a multi-class SVM is trained, how much improvement can be expected?#
We chose DCNNs because they are very efficient and have features sharing. We did not implement our model replacing DCNNs by HOG+SVMs, so we cannot be sure what the results would be. But our paper shows that we compare favorably with other models that use HOG+SVMs.

#All the examples shown in the left most of Figure 1 seem to be from the frontal view. In this work, has the variation of relative position w.r.t viewpoint been explicitly handled?#
Figure 1 was kept simple to illustrate the basic ideas. Our model allows several types of spatial relationships between parts, which gives our model the ability to handle implicitly variation of relative position caused by pose or viewpoint. These types of relationships are learned by clustering.

@R31:
#conditioning of pairwise terms in a graphical model for human pose estimation has been done before by [SJT10] and [11] … Moreover, it is not clear that local conditioning of pairwise terms has more advantages over global conditioning …#
Yes, we should refer to [SJT10,11] and CRF literature on data-dependent priors. The reviewer is right that adding non-local information would probably improve performance. In this paper we chose to make the model as simple and efficient as possible. But we will explore non-local cues in our future work.

#the proposed method heavily builds on Y&R model [20,21] and reads like an extension of Y&R model with image-conditioned pairwise terms.#
We are building on previous computer vision work on Pictorial Structure models including [20,21]. Our work is a combination of Pictorial models for structure, data-dependent priors for spatial relationships, and DCNNs for parts (including relationships). Our contribution is to show when we combine these ingredients together, we get good results.

#choosing a number of mixture components for the pairwise terms (11) is not justified. Y&R model [21] uses 6 components only.#
The word “mixture” is used in two different senses. The mixture components in our model corresponds to different types of pairwise spatial relationships. But in [21], mixture components correspond to different HOG templates of each part (see discussion in Section 2.2). They have tested their model, although on the Parse dataset, with more than 6 components (up to 11) and their performance seems to saturate at 6, see [21](Fig. 7).

#using 0-1 loss during training is suboptimal. A more proper way would be to use the same loss …#
Yes, we used the 0-1 loss for simplicity. We expect to improve if we use the same loss.

#comparison to Deep Pose [17] on LSP dataset is not completely fair, as [17] report results using person-centric annotations#
We cannot have completely fair comparison with [17] on the LSP dataset, since they use 10 times more training images. All the other methods we compared with on the LSP dataset use the observer-centric annotations, like us.

On the FLIC and Buffy dataset, our settings with [17] are the same, and we show significant improvement.

#some citations are missing#
Sorry, we will add them.

#unrealistic poses resulting from data augmentation … may hurt the performance … it has been suggested [20,21] that the appearance of body parts is rotation dependent …#
We rotate the images to get more examples of body parts with different orientations. The image rotation does NOT preserve the pairwise type labels of each part. When the image is rotated, the appearance of body part changes, and so does its pairwise spatial relationships with its neighbors. See the elbows along the diagonal of the left Panel in Figure 1.

It's our motivation that appearance (local image measurements) of body part is related to its spatial relationships with its neighbors, and thus can be useful to predict the spatial relationships.

#what is the training/testing time of the method?#
With our unoptimized code, training is done in around two days, and testing is around 6 seconds per image. We use Nvidia GTX 780 GPU and a single CPU core.